# Predictors of Hypertension in Mauritians with Normotension and Prehypertension at Baseline: A Cohort Study

**DOI:** 10.3390/ijerph15071394

**Published:** 2018-07-02

**Authors:** Sudhirsen Kowlessur, Zhibin Hu, Jaysing Heecharan, Jianming Wang, Juncheng Dai, Jaakko O. Tuomilehto, Stefan Söderberg, Paul Zimmet, Noël C. Barengo

**Affiliations:** 1Department of Epidemiology, School of Public Health, Nanjing Medical University, Nanjing 211166, China; sukowlessur@govmu.org (S.K.); jmwang@njmu.edu.cn (J.W.); djc@njmu.edu.cn (J.D.); 2Ministry of Health and Quality of Life, Port Louis 11321, Mauritius; jheecharan@govmu.org; 3Department of Public Health, University of Helsinki, 00100 Helsinki, Finland; jaakko.tuomilehto@helsinki.fi; 4Department of Public Health and Clinical Medicine, Umeå University, SE-901 87 Umea, Sweden; stefan.soderberg@umu.se; 5Department of Medicine, Monash University, Melbourne, VIC 3800, Australia; paul.zimmet@monash.edu; 6Medical and Population Health Sciences Research, Herbert Wertheim College of Medicine, Florida International University, Miami, FL 33199, USA; nbarengo@fiu.edu

**Keywords:** systolic blood pressure, diastolic blood pressure, ethnicity, follow-up, risk prediction, central obesity, overweight, Africa

## Abstract

Information on the predictors of future hypertension in Mauritians with prehypertension is scant. The aim of this study was to analyze the 5-year and 11-year risk of hypertension and its predictors in people with normotension and prehypertension at baseline in Mauritius in 1987. This was a retrospective cohort study of 883 men and 1194 women of Mauritian Indian and Mauritian Creole ethnicity, aged 25–74 years old, free of hypertension at baseline in 1987 with follow-up examinations in 1992 and 1998 using the same methodology. The main outcome was 5- and 11-year risk of hypertension. Odds ratios (OR) and corresponding 95% confidence intervals (CI) were calculated. The 5-year risk of hypertension was 5.4-times higher in people with prehypertension compared with normotensive individuals at baseline. The corresponding odds for prehypertensive people at baseline regarding 11-year hypertension risk was 3.39 (95% CI 2.67–4.29) in the adjusted logistic regression models. Being of Creole ethnicity (OR 1.42; 95% CI 1.09–1.86) increased the 11-year odds of hypertension compared with the Indian population. It is of importance to screen for people with prehypertension and implement strategies to reduce their systolic blood pressure levels to the recommended levels of 120/80 mmHg. Special attention needs to be given to Mauritians of Creole ethnicity.

## 1. Introduction

There is sufficient scientific evidence to conclude that hypertension is a major global public health problem, as increased blood pressure is the most important risk factor for cardiovascular diseases (CVD), including stroke [1,2,3,4]. Furthermore, hypertension is one of the leading causes of the global burden of disease, accounting for 212 million global disability-adjusted life-years in 2015 [5]. More than a decade ago, the seventh report of the Joint National Committee on Prevention, Detection, Evaluation, and Treatment of High Blood Pressure (JNC 7) proposed a new blood pressure category of 120–139 mmHg systolic blood pressure (SBP) or 80–89 mmHg diastolic blood pressure, referring to it as prehypertension [6]. This proposal was based on a meta-analysis of 61 prospective studies revealing that mortality from ischemic heart disease and stroke increases in a log-linear relationship with blood pressure (BP) [7,8]. Globally, it has been estimated that the prevalence of prehypertension may be as high as 50% in the middle-aged population [9,10].

Previous studies have reported that in people with BP 130–139 mmHg systolic and/or 85–89 mmHg diastolic blood pressure, the risk of developing hypertension is threefold that of normotensives with BP <120/<80 mmHg [11,12]. Even though some studies have assessed the risk factors of short- and long-term risk of hypertension in the Mauritian population [13,14], no information is available on the risk of hypertension in people with prehypertension at baseline. Moreover, information on predictors of future risk of hypertension is scant.

The aim of this study was to analyze the 5-year and 11-year risk of hypertension and its predictors in people with normotension and prehypertension at baseline in Mauritius in 1987.

## 2. Materials and Methods 

### 2.1. Study Population and Selection

Mauritius is located in the southwestern Indian Ocean. The multiethnic population consists of 68% individuals of Asian Indian origin (52% Hindu and 16% Muslim), 3% individuals of Chinese origin, and 27% of mixed African and Malagasy ancestry with some European and Indian admixture (Creoles) [15].

A population-based survey was undertaken in 10 locations in 1987 and the age range of the subjects was 25–74 years old. In 1987, 10 randomly selected (with probability proportional to size) population clusters and a purposely selected area of Chinatown in the capital, Port Louis, were surveyed, and all eligible residents were invited to participate. In 1992 and 1998, the same clusters were resurveyed, as well as an additional three clusters selected to assess if trends in disease and risk-factor distribution observed in the original study cohort also occurred in independent clusters [16]. In 1998, the Chinatown cluster was not surveyed. In both 1992 and 1998, all previous participants were invited plus all other current eligible residents in each cluster. The numbers of participants were 5083 (86% response rate), 6616 (90% response rate), and 6291 (87% response rate), respectively, in each survey. Altogether, 9559 individuals participated and 27% took part in all three surveys. Details of the survey methodology and descriptions of the population clusters have been published previously [15,16,17,18,19].

The inclusion criteria for the current cohort study were as follows: (i) individuals who participated in all three surveys; (ii) age range 25–74 years old and (iii) living in Mauritius. People who had hypertension at baseline or with missing data were excluded from the analysis. The final sample size of the follow-up cohort consisted of 883 men and 1194 women.

### 2.2. Baseline Assessment of Hypertension Risk Factors

The survey methodology was the same in all three surveys. All eligible adults were asked to attend a survey site between 8.00 and 10.00 a.m. after an overnight fast. After registration, trained local nurses administered a questionnaire and anthropometric measurements were done [15,16,17]. The questionnaire included standardized questions on health behavior, including smoking habits, and other health related and sociodemographic factors.

In each survey, BP was measured twice on the right arm of the participant with elbow level at heart after sitting for 5–10 min using a standard mercury sphygmomanometer. The mean of the two measurements was used to define hypertension incidence. The recommendations of the World Health Organization Multinational Monitoring of Trends and Determinants of CVD (MONICA) project were followed at each survey [20,21]. Prior to each survey, the examiners received detailed training in the same way on how to measure the BP, following the MONICA project recommendations. The measurement qualification was then confirmed by the person responsible for training using the double-stethoscope method. People with systolic blood pressure <120 mmHg and diastolic blood pressure <80 mmHg were considered as normotensive. A systolic blood pressure of 120–139 mmHg or a diastolic blood pressure of between 80 and 89 mmHg in people who were free of antihypertensive medication was considered as prehypertension. Hypertension was defined as systolic blood pressure of at least 140 mmHg and/or diastolic blood pressure of at least 90 mmHg according to the 1999 World Health Organization-International Society of Hypertension guidelines and/or on hypertension treatment [22]. Individuals who developed hypertension during the follow-up were counted as incident cases.

Waist circumference was measured twice at the midpoint between the lower margin of the ribs and the iliac crest to the nearest 0.5 cm. The third measurement was taken if the first two readings differed by more than 2 cm. The mean of the closest two measurements was used to calculate waist circumference. Waist circumference was then dichotomized into normal (men <90 cm and women <80 cm) and increased (men >90 cm and women >80 cm).

Body Mass Index (BMI) was calculated as weight in kilograms divided by the square of height in meters (kg/m^2^). According to ethnicity, study participants were then categorized into four categories as follows: (i) <18.5 kg/m^2^; (ii) 18.5–24.9/18.5–22.9 kg/m^2^; (iii) 25–29.9/23–29.9 kg/m^2^; and (iv) >30 kg/m^2^.

Physical activity was determined by a self-administered questionnaire following the recommendations of WHO MONICA project protocol [18,21]. Both leisure-time and occupational physical activity were categorized on a four-level scale (sedentary, light, moderate, and heavy) based on usual patterns over the past year. Interviewers were given guidelines for allocating activity levels for both leisure-time and occupational physical activity scales. Occupational physical activity was classified as sedentary for office workers and the unemployed, light for shop assistants and general housework, moderate for trade workers such as carpenters, and heavy for building laborers and sugarcane cutters. Leisure-time physical activity was graded as sedentary for those generally housebound with no regular outside activity, light for regular but relaxed pursuits such as gardening and walking, moderate for active sports such as jogging, volleyball, or cycling for ≥ 30 min 1–2 days/week, and heavy for active sports undertaken ≥ 3 days/week. 

For the statistical analysis, moderate and heavy leisure-time physical activity categories were combined.

Smoking status (yes/no) was determined by self-report and never-smokers were used as the reference category. Ethnicity was determined by self-report and divided into (i) Indian, (ii) Creole, or (iii) Chinese.

Educational level was categorized into four categories: (i) None; (ii) 1–6 years; (iii) 7–12 years; and (iv) >12 years.

All study participants not taking diabetes medication had a 2-h 75-g oral glucose (glucose monohydrate) tolerance test (OGTT) [15]. Fluoridated tubes were used for fasting and 2-h venous blood samples, which were centrifuged (nonrefrigerated centrifuges) and separated within 20 min. Plasma glucose was measured on-site in 1987 and 1992 using YSI glucose analyzers (Yellow Springs Instruments, Yellow Springs, OH, USA) within 3 h of collection with quality controls measured several months later in Newcastle upon Tyne, UK, at the reference laboratory, which is a member of the Wellcome quality assurance scheme. This showed a small but consistent and systematic fall in glucose over time. In 1998, plasma was frozen immediately, and glucose was measured approximately four months later in Newcastle upon Tyne, UK, using the same technology. Considering the delay in analysis, the 1998 values were adjusted upwards using an equation (adjusted glucose = 0.0288 + 1.037 × measured glucose) based on the difference between on-site values and quality controls from the 1987 and 1992 surveys. Glucose tolerance status was determined according to 1999 WHO criteria [23]. Impaired fasting glucose (IFG) was defined as fasting plasma glucose concentration of more than 6.0 and less than 7.0 mmol/L. Impaired glucose tolerance (IGT) was defined as glucose plasma concentration of more than 7.80 to less than 11.1 mmol/L after OGTT. Diabetes mellitus (DM2) was defined as fasting glucose concentration from 7.0 mmol/L or from 11.1 mmol/L 2 h after OGTT. Diabetes was diagnosed if participants reported a history of diabetes and were taking hypoglycemic medication, or the fasting plasma glucose level was ≥7.0 mmol/L and/or the 2-h value was ≥11.1 mmol/L.

### 2.3. Statistical Analysis

All analyses were performed using IBM SPSS Statistics version 20 (IBM Corp. Released 2011. IBM SPSS Statistics for Windows, Version 20.0. Armonk, NY, USA). Categorical variables are presented as proportions. Collinearity diagnostics were performed prior to the logistic regression analysis to check for multicollinearity. Adjusted and unadjusted logistic regression analysis were used to test for the associations between predictors and 5-year and 11-year risk of hypertension, respectively. Odds ratios (OR) and their corresponding 95% confidence intervals (CI) are presented. Forward logistic regression models were used for the adjusted models. The covariates were chosen after revising scientific evidence on risk factors for hypertension. The covariates were entered into the model according to the probability of the likelihood-ratio statistic based on conditional parameter estimates. As no evidence of bias or interaction was found by participants’ sex, the data were analyzed combining men and women. The odds ratios and the respective 95% confidence intervals were calculated. A *p*-value of <0.05 was considered as a level of statistical significance. The goodness-of-fit of the statistical models was assessed using the Hosmer Lemeshow test.

### 2.4. Ethical Considerations

This study followed the Good Clinical Practice guidelines and the guidelines of the Helsinki Declaration. The study of each survey (1987, 1992, 1998) was approved by the Ethics Committee of the International Diabetes Institute. The survey protocol was also reviewed and approved by the Alfred Healthcare Group Ethics Committee (Melbourne, Australia) as well as the National Ethics Committee of Mauritius. All study participants gave their written informed consent prior to the participation in the study

## 3. Results

### 3.1. Baseline Characteristics of the Study Participants

The characteristics of the study participants according to baseline blood pressure category are shown in Table 1.

The number of people with normotension at baseline was 1034 (64% women), whereas 1043 people had prehypertension in 1987 (51% women). A higher percentage of individuals with prehypertension seem to be of Creole ethnicity.

### 3.2. Predictors of 5- and 11-Year Risk of Hypertension in Normotensive People at Baseline

In the unadjusted analysis, age was statistically significantly associated with 5-year risk of hypertension in normotensive Mauritians at baseline (Table 2). People 45–54 years-of-age had a 4.68-fold increased 5-year risk of hypertension (95% CI 2.05–10.70), whereas the corresponding odds for those 55–64 years old was 4.59 (95% CI 1.53–13.78). In the stepwise regression model for 5-year risk, the only predictors that remained in the model were sex and age groups. Women with normotension at baseline had a 31% reduced odds of hypertension (95% CI 0.57–0.85) compared with men after adjustment for age. Several predictors were identified, however, as predictors of hypertension 11-years after baseline examination in the adjusted models. Besides age group, occupational physical activity decreased the risk of hypertension, whereas being overweight increased it. Mauritians with light occupational physical activity had an odds ratio of 0.29 (95% CI 0.15–0.55) of hypertension after 11 years of follow-up. The corresponding risk reduction for those with moderate occupational physical activity was 62% (95% CI 0.18–0.79). People who were overweight had an odds ratio for 11-year hypertension of 2.82 (95% CI 1.84–4.33). The risk increase for those with obesity was not statistically significant (OR 2.07; 95% CI 0.84–5.11).

### 3.3. Predictors of 5- and 11-Year Risk of Hypertension in Prehypertensive People at Baseline

Only baseline age group was statistically significantly associated with 5-year risk of hypertension in Mauritians with prehypertension at baseline in the adjusted logistic regression models (Table 3). Whereas the risk increases for those 35–44 years-of-age was 3.27 (95% CI 1.97–5.42), the 45–54 years-old Mauritians had an odds ratio of 4.64 (95% CI 2.73–7.92) of hypertension after the 11-year follow-up. The corresponding odds ratios for those 55–64 years old at baseline and >65 years old were 7.5 (95% CI 4.22–13.32) and 10.98 (95% CI 4.82–25.05), respectively. Age, ethnicity, and being overweight were associated with increased odds of hypertension after 11 years of follow-up. While the Creole population had a 54% (95% CI 1.13–2.09) risk increase compared with the Indian population, being of Chinese ethnicity did not statistically significantly increase the risk of hypertension. Finally, being overweight was associated with a 1.46-fold increased likelihood of being diagnosed with hypertension after 11 years (95% CI 1.09–1.97) in the stepwise regression models.

### 3.4. Adjusted Predictors of 5- and 11-Year Risk of Hypertension in Normotensive and Prehypertensive People at Baseline in Mauritius in 1987

Table 4 shows the adjusted predictors of 5- and 11-year risk of hypertension in normotensive and prehypertensive people at baseline in Mauritius in 1987. The 5-year risk of hypertension was 5.4 times higher in people with prehypertension compared with normotensive individuals at baseline. The corresponding odds for prehypertensive people at baseline regarding 11-year hypertension risk was 3.39 (95% CI 2.67–4.29) in the adjusted logistic regression models. Age was a statistically significant predictor for hypertension after 5 and 11 years of follow-up. Being of Creole ethnicity (OR 1.42; 95% CI 1.09–1.86) increased the 11-year odds of hypertension compared with the Indian population. Whereas light (OR 0.52; 95% CI 0.34–0.77), moderate (OR 0.51; 95% CI 0.33–0.80), and heavy (OR 0.59; 95% CI 0.37–0.93) occupational physical activity reduced the 11-year risk of hypertension, overweight (OR 1.83; 95% CI 1.43–2.34) and obesity (OR 1.8; 95% CI 1.16–2.81) increased it.

## 4. Discussion

Our study revealed that Mauritians with prehypertension have a three- to five-fold increased risk of future hypertension compared with normotensive people after adjustment for age, ethnicity, BMI, and occupational physical activity. In people with normotension at baseline, age, occupational physical activity, and being overweight were independent predictors of 11-year risk of hypertension. The corresponding predictors of hypertension after an 11-year follow-up for prehypertensive people at baseline were age, being of Creole ethnicity, and being overweight.

Our results are consistent with findings of previous studies revealing that people with prehypertension are more likely to become hypertensive over time, especially if they are of African-American descent [24,25]. In a prospective cohort study of 18,865 non-hypertensive persons, baseline systolic blood pressure between 130 and 139 mmHg (Hazard Ratio (HR) 1.77) and between 120 and 129 mmHg (HR 1.52) were the strongest statistically significant predictors of hypertension [25]. Furthermore, in people with BP 130–139 mmHg systolic and/or 85–89 mmHg diastolic blood pressure, the risk of developing hypertension is threefold that of normotensives with BP <120/<80 mmHg [11,12]. These estimates are very similar to the ones we found in our study population regarding the 11-year risk of hypertension in people with prehypertension. It has to be kept in mind, though, that there is substantial heterogeneity within the category of prehypertension [8]. For instance, the risk of progressing to hypertension and developing CVD is higher in individuals with BP 130–139/85–89 mmHg than in those with BP 120–129/80–84 mmHg [9,26].

Current scientific evidence in different populations has shown that moderate to high levels of leisure-time physical activity reduce the risk of future hypertension [27,28]. Some of the proposed mechanisms of how chronic exercise decreases blood pressure include a decrease in systemic vascular resistance caused by decreased sympathetic activity, reduced plasma renin activity, decreased concentrations of catecholamines, increased urinary sodium excretion, and insulin sensitivity [29,30]. In addition, exercise training has been shown to improve endothelium-dependent vasodilatation, both in epicardial vessels and in resistance vessels, in patients with coronary artery disease [30]. In the Mauritius population, the prevalence of moderate or high physical activity at baseline was less than 10%. Furthermore, 6 out of 10 people were sedentary during leisure time at baseline. Thus, the statistical power of our study may not be sufficient to show any protective effect of moderate to high leisure-time physical activity on future hypertension as indicated by the large confidence intervals of our results. Evidence for the association between occupational physical activity is less consistent [27,28,31,32,33,34]. In agreement with a recent study, our findings revealed a protective effect of occupational physical activity on future risk of hypertension [32]. However, other studies have failed to show an association between occupational physical activity and risk of hypertension [27,28,34]. These inconsistent findings have been commonly explained by the confounding effect of muscle mass on BMI in physically intense workers, the small estimated effect that occupational physical activity has on cardiorespiratory fitness, and the inflammatory process involved in the prolonged elevated heart rate during hard work [35,36].

Limited information exists on the association between baseline blood pressure and future hypertension in people of Creole ethnicity. Current scientific evidence has shown that Creole ethnicity is associated with a higher prevalence of hypertension compared with other populations [37]. Hypertension was the single cardiovascular risk factor associated with small vessel infarction in Haitian immigrants in United States (US)-based studies [37,38,39,40]. Thus, investigation of factors influencing this condition among this population group is important. In disagreement with a previous study, Creole ethnicity was associated with the long-term risk of hypertension compared with the population of Indian or Chinese origin [13]. This may be due to the fact that our study followed up the same population cohort during the entire follow-up process, unlike a previous study in the same population [13]. Our study is one of the very few existing cohort studies that assessed the risk factors of future hypertension in the same cohort using measurements at three different time points. Previous studies in the same populations did not follow up the same people over time [13,14]. Furthermore, most previous studies conducted in Creole populations were cross-sectional in nature [38,39,40]. The reasons for the increased risk of hypertension in the Creole population are poorly understood but may include genetic factors and different dietary habits.

The main limitations of our study are those in epidemiological studies in general. The baseline assessment of our cohort was limited to an examination on a single day when participants were enrolled in the study, which is common in large cohort studies. The study design cannot account for changes in lifestyle or risk factors of hypertension leading to a shift of participants between categories during the study period. Moreover, we were not able to analyze the data using survival analysis, as information on the individual times-to-event was not available for the study participants who developed the outcome event. As the follow-up time of every study participant was identical, Cox regression analysis (which takes into account variable follow-up times) would not have been an appropriate method. All study participants were re-examined after 5 and 11 years at the same time point. Thus, if they had hypertension at the follow-up examination, we would not know at which time point they developed hypertension. Additionally, using the same follow-up time for Kaplan–Meier survival time would give inaccurate data on survival times for people who developed hypertension. Furthermore, follow-up times would be the same for each individual and lead to similar estimated odds ratios. Therefore, we decided to use logistics regression models to calculate the point estimates for the associations. Even though our analyses were adjusted for most lifestyle factors, residual confounding due to other lifestyle factors such as unmeasured dietary habits cannot be excluded. This may influence the association between baseline blood pressure levels and future hypertension. However, as this may be the fact for exposed and unexposed people, this most likely present a nondifferential misclassification bias.

## 5. Conclusions

In conclusion, as Mauritians with prehypertension have a several times increased risk of future hypertension already within five years, it is of importance to screen for people with prehypertension and implement strategies to reduce their systolic blood pressure levels to the recommended levels of 120/80 mmHg. This may be achieved by pharmaceutical therapy or changes in lifestyle habits, such as reducing overweight/obesity and increasing physical activity. A cornerstone to avoid future hypertension is the primary health-care system and the systematic implementation of evidence-based clinical guidelines in the control of hypertension. Special attention needs to be given to Mauritians of Creole ethnicity as they have an increased risk of future hypertension compared to other ethnicities. Future studies may assess in more detail dietary factors and the specific reasons why the Creole population has an increased risk of hypertension.

## Figures and Tables

**Table 1 ijerph-15-01394-t001:** Characteristics of the study participants according to baseline blood pressure category.

	Normotensive		Prehypertensive	
	(*n* = 1034)		(*n* = 1043)	
	%	(*n*)	%	(*n*)
**Sex**				
Men	36	(368)	49	(515)
Women	64	(666)	51	(528)
**Age**				
<35 years old	47	(486)	38	(395)
35–45 years old	33	(345)	30	(316)
45–55 years old	13	(133)	19	(191)
55–65 years old	5	(52)	11	(110)
>65 years old	2	(18)	3	(31)
**Educational level**				
None	16	(161)	17	(172)
1–6 years	49	(507)	50	(524)
7–12 years	31	(320)	30	(310)
>12 years	4	(46)	3	(37)
**Ethnic group**				
Indian	81	(841)	69	(715)
Creole	16	(160)	26	(270)
Chinese	3	(33)	5	(58)
**Current smoking**				
No	74	(760)	73	(755)
Yes	26	(273)	27	(287)
**Occupational physical activity**				
Low	8	(81)	7	(77)
Light	57	(586)	51	(532)
Moderate	19	(201)	22	(228)
Heavy	16	(166)	20	(206)
**Leisure-time physical activity**				
Low	63	(655)	53	(556)
Light	29	(302)	36	(379)
Moderate or heavy	8	(77)	11	(108)
**BMI (according to ethnicity)**				
<18.5 kg/m^2^	12	(119)	7	(73)
18.5–24.9/18.5–22.9 kg/m^2^	47	(480)	39	(400)
25–29.9/23–29.9 kg/m^2^	37	(370)	46	(480)
>30 kg/m^2^	4	(44)	8	(81)
**Waist**				
Normal	82	(829)	74	(763)
Increased	18	(183)	26	(273)
**Glucose metabolism**				
Normal	80	(820)	69	(713)
Impaired fasting glucose	3	(27)	4	(39)
Impaired glucose tolerance	13	(133)	17	(171)
Diabetes mellitus	4	(45)	11	(115)

BMI—body mass index.

**Table 2 ijerph-15-01394-t002:** Predictors of 5- and 11-year risk of hypertension in normotensive people at baseline in Mauritius in 1987.

	5-Year Odds		5-Year Odds		11-Year Odds		11-Year Odds	
	Unadjusted		Adjusted		Unadjusted		Adjusted	
	OR	(95% CI)	OR	(95% CI)	OR	(95% CI)	OR	(95% CI)
**Sex**								
Men	Ref		Ref		Ref			
Women	0.84	(0.43–1.06)	0.69	(0.57–0.85)	0.94	(0.64–1.37)		
**Age**								
<35 years old	Ref		Ref		Ref		Ref	
35–45 years old	1.16	(0.47–2.82)	0.99	(0.39–2.48)	1.72	(1.09–2.71)	1.37	(0.85–2.21)
45–55 years old	4.68	(2.05–10.70)	4.13	(1.78–9.58)	3.63	(2.17–6.08)	3.22	(1.86–5.58)
55–65 years old	4.59	(1.53–13.78)	4.35	(1.45–13.06)	2.73	(1.27–5.86)	3.09	(1.38–6.88)
>65 years old	na	na	na	na	3.28	(1.03–10.43)	3.11	(0.93–10.39)
**Ethnic group**								
Indian	Ref				Ref			
Creole	1.28	(0.55–2.98)			0.81	(0.47–1.39)		
Chinese	1.81	(0.41–7.91)			1.21	(0.46–3.21)		
**Educational level**								
None	Ref				Ref			
1–6 years	0.79	(0.34–1.82)			0.81	(0.50–1.31)		
7–12 years	0.49	(0.18–1.33)			0.46	(0.26–0.81)		
>12 years	0.87	(0.18–4.24)			0.61	(0.22–1.67)		
**Current smoking**								
No	Ref				Ref			
Yes	1.19	(0.58–2.43)			0.82	(0.53–1.27)		
**Occupational physical activity**								
Low	Ref				Ref		Ref	
Light	0.92	(0.27–3.16)			0.44	(0.24–0.78)	0.29	(0.15–0.55)
Moderate	1.51	(0.41–5.54)			0.45	(0.23–0.89)	0.38	(0.18–0.79)
Heavy	0.64	(0.14–2.94)			0.59	(0.30–1.17)	0.52	(0.24–1.06)
**Leisure-time physical activity**								
Low	Ref				Ref			
Light	1.18	(0.59–2.36)			1.45	(0.98–2.15)		
Moderate or heavy	0.35	(0.05–2.59)			1.04	(0.50–2.17)		
**BMI (according to ethnicity)**								
<18.5 kg/m^2^	na	na			0.17	(0.04–0.71)	0.16	(0.04–0.66)
18.5–24.9/18.5–22.9 kg/m^2^	Ref				Ref		Ref	
25–29.9/23–29.9 kg/m^2^	1.17	(0.60–2.28)			2.44	(1.63–3.64)	2.82	(1.84–4.33)
>30 kg/m^2^	0.56	(0.07–4.32)			2.2	(0.96–5.03)	2.07	(0.84–5.11)
**Waist**								
Normal	Ref				Ref			
Increased	1.26	(0.57–2.81)			1.99	(1.30–3.04)		
**Glucose metabolism**								
Normal	Ref				Ref			
Impaired fasting glucose	2.54	(0.57–11.34)			1.34	(0.46–3.97)		
Impaired glucose tolerance	2.04	(0.89–4.61)			1.53	(0.92–2.54)		
Diabetes mellitus	2.27	(0.66–7.83)			1.93	(0.90–4.13)		

OR—odds ratio; CI—confidence interval; na—not applicable; Ref—reference group.

**Table 3 ijerph-15-01394-t003:** Predictors of 5- and 11-year risk of hypertension in prehypertensive people at baseline in Mauritius in 1987.

	5-Year Odds		5-Year Odds		11-Year Odds		11-Year Odds	
	Unadjusted		Adjusted		Unadjusted		Adjusted	
	OR	(95% CI)	OR	(95% CI)	OR	(95% CI)	OR	(95% CI)
**Sex**								
Men	Ref				Ref			
Women	1.25	(0.90–1.73)			1.21	(0.94–1.56)		
**Age**								
<35 years old	Ref		Ref		Ref		Ref	
35–45 years old	3.33	(2.01–5.51)	3.27	(1.97–5.42)	1.88	(1.36––2.60)	1.78	(1.28–2.48)
45–55 years old	4.77	(2.80–8.10)	4.65	(2.73–7.92)	2.7	(1.87–3.89)	2.57	(1.77–3.74)
55–65 years old	7.52	(4.24–13.35)	7.5	(4.22–13.32)	4.56	(2.92–7.12)	4.44	(2.82–7.01)
>65 years old	11.16	(4.90–25.46)	10.98	(4.82–25.05)	3.37	(1.61–7.07)	3.61	(1.69–7.69)
**Ethnic group**								
Indian	Ref				Ref		Ref	
Creole	1.21	(0.84–1.75)			1.35	(1.01–1.80)	1.54	(1.13–2.09)
Chinese	1.25	(0.63–2.48)			1.35	(0.78–2.32)	1.16	(0.65–2.06)
**Educational level**								
None	Ref				Ref			
1–6 years	0.66	(0.44–1.01)			0.64	(0.45–0.91)		
7–12 years	0.47	(0.29–0.77)			0.58	(0.40–0.85)		
>12 years	0.28	(0.08–0.97)			0.43	(0.19–0.93)		
**Current smoking**								
No	Ref				Ref			
Yes	1.04	(0.72–1.49)			1.04	(0.79–1.38)		
**Occupational physical activity**								
Sedentary	Ref				Ref			
Light	1.03	(0.55–1.91)			0.75	(0.47–1.22)		
Moderate	0.79	(0.40–1.56)			0.55	(0.33–0.94)		
Heavy	0.68	(0.34–1.38)			0.62	(0.36–1.05)		
**Leisure-time physical activity**								
Low	Ref				Ref			
Light	0.87	(0.62–1.23)			0.75	(0.57–0.99)		
Moderate or heavy	0.45	(0.23–0.90)			0.54	(0.34–0.85)		
**BMI (according to ethnicity)**								
<18.5 kg/m^2^	1.77	(0.93–3.35)			0.73	(0.42–1.28)	0.69	(0.38–1.25)
18.5–24.9/18.5–22.9 kg/m^2^	Ref				Ref		Ref	
25–29.9/23–29.9 kg/m^2^	1.56	(1.07–2.26)			1.4	(1.06–1.84)	1.46	(1.09–1.97)
>30 kg/m^2^	2.1	(1.16–3.79)			1.84	(1.13–2.98)	1.63	(0.98–2.70)
**Waist**								
Normal	Ref				Ref			
Increased	1.24	(0.86–1.77)			1.48	(1.11–1.96)		
**Glucose metabolism**								
Normal	Ref				Ref			
Impaired fasting glucose	0.69	(0.24–1.97)			1.01	(0.51–2.00)		
Impaired glucose tolerance	1.6	(1.05–2.44)			1.58	(1.12–2.22)		
Diabetes mellitus	2.11	(1.33–3.37)			1.85	(1.25–2.76)		

**Table 4 ijerph-15-01394-t004:** Adjusted predictors of 5- and 11-year risk of hypertension in normotensive and prehypertensive people at baseline in Mauritius in 1987.

	5-Year Odds		11-Year Odds	
	Adjusted		Adjusted	
	OR	(95% CI)	OR	(95% CI)
**Blood pressure at baseline**				
Normotensive	Ref		Ref	
Prehypertensive	5.4	(3.73–7.82)	3.39	(2.67–4.29)
**Age**				
<35 years old	Ref		Ref	
35–45 years old	2.49	(1.61–3.83)	1.65	(1.26–2.17)
45–55 years old	4.38	(2.79–6.89)	2.74	(2.01–3.75)
55–65 years old	6.45	(3.91–10.65)	3.81	(2.58–5.62)
>65 years old	7.27	(3.46–15.29)	3.31	(1.74–6.30)
**Ethnic group**				
Indian			Ref	
Creole			1.42	(1.09–1.86)
Chinese			1.08	(0.65–1.80)
**Occupational physical activity**				
Sedentary			Ref	
Light			0.52	(0.34–0.77)
Moderate			0.51	(0.33–0.80)
Heavy			0.59	(0.37–0.93)
**BMI (according to ethnicity)**				
<18.5 kg/m^2^			0.51	(0.38–1.25)
18.5–24.9/18.5–22.9 kg/m^2^			Ref	
25–29.9/23–29.9 kg/m^2^			1.83	(1.43–2.34)
>30 kg/m^2^			1.8	(1.16–2.81)

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
