# Peer review of "Predictors of Hypertension in Mauritians with Normotension and Prehypertension at Baseline: A Cohort Study"

_ijerph, 2018, doi:10.3390/ijerph15071394_

Round 1
Reviewer 1 Report
The methodology of physical activity (PA) assessment should be clarified and the criteria for classifying into low, medium and high PA should be presented.
The main limitation of the manuscript is inappropriate statistical analysis. The authors emphasize that this is a cohort study, present risks and predictors of hypertension but they do not use survival analysis. Why? Only odds ratios at different times are presented while they are not taking into account individual time of follow-up.
Analysis should be repeated using Kaplan-Meier survival tables/survival probability plots and Cox regression for adjusting RRs for covariates. Study sample could be analyzed in separate subgroups of normotensives and prehypertensives, in men and women separately due to possible interaction of gender on RRs, also, if authors prefer, RRs could be presented in different periods of follow-up (eg. after 5, 11 years…). Incident hypertension could be classified as cohort event, whereas study sample could be divided into more levels of BP to assess the dose-response effect or just the baseline systolic and diastolic BP could be entered as predictive factor for developing hypertension. Please update your analyses with methods suitable for cohort with long term of follow-up!
Author Response
Response to Reviewer’s comments
Manuscript ID: ijerph-304588
Editor’s and Reviewer’s Comments
| Our Response
| Location of edits |
Reviewer #1 | ||
The methodology of physical activity (PA) assessment should be clarified and the criteria for classifying into low, medium and high PA should be presented. | We have provided more information about the assessment of physical activity at baseline in the methodology section. The description reads now as follows:
Physical activity was determined by a self-administered questionnaire following the recommendations of World Health Organization Multinational Monitoring of Trends and Determinants of CVD (WHO MONICA) project protocol [21,23]. Both leisure time and occupational physical activity were categorized on a four-level scale (sedentary, light, moderate, and heavy) based on usual patterns over the past year. Interviewers were given guidelines for allocating activity levels for both leisure time and occupational physical activity scales. Occupational physical activity was classified as sedentary for office workers and the unemployed, light for shop assistants and general housework, moderate for trade workers such as carpenters, and heavy for building laborers and sugarcane cutters. Leisure time physical activity was graded as sedentary for those generally housebound with no regular outside activity; light for regular but relaxed pursuits such as gardening and walking; moderate for active sports such as jogging, volleyball, or cycling for >=30 min 1-2 days/week; and heavy for active sports undertaken >+ 3 days/week.
Following reference has been added:
Dowse, G.K.; Zimmet, P.Z.; Gareeboo, H.; Alberti, G. M.M.; Tuomilehto, J.; Finch, C.F.; Chitson, P.; Tulsidas, H. Abdominal obesity and physical inactivity as risk factors for NIDDM and impaired glucose tolerance in Indian, Creole, and Chinese Mauritians. Diabetes Care 1991, 14(4), 271-82. | Page 3, line 122. |
The main limitation of the manuscript is inappropriate statistical analysis. The authors emphasize that this is a cohort study, present risks and predictors of hypertension but they do not use survival analysis. Why? Only odds ratios at different times are presented while they are not taking into account individual time of follow-up | Our study is a cohort study following up individuals free of hypertension at baseline with the outcome of incident hypertension. As we do not have follow-up time of the study participants, we were not able to use Cox regression models or present survival curves. All stud participants were re-examined after 5- and 11 –years at the same time point. Thus, if they had hypertension at the follow-up examination, we would not know at which time point they developed hypertension. Using the same follow-up time (5 yrs, respectively 11 yrs) for survival time would inaccurate survival times for people who developed hypertension. Furthermore, follow-up times will be the same and lead to similar estimated as odds ratios. Therefore, we decided to use logistics regression models, a perfectly valid statistical method to calculate associations if survival times are not available.
We have addressed this in the limitations of the study section adding the following text:
“Moreover, we were not able to analyze the data using survival analysis as information on the outcomes were only assessed at the 5-year and 11-year examination visits, and individual times-to-event information was not available for the study participants who developed the outcome event.” | Page 12, line 298 |
Analysis should be repeated using Kaplan-Meier survival tables/survival probability plots and Cox regression for adjusting RRs for covariates. Study sample could be analyzed in separate subgroups of normotensives and pre-hypertensives, in men and women separately due to possible interaction of gender on RRs, also, if authors prefer, RRs could be presented in different periods of follow-up (eg. after 5, 11 years…). Incident hypertension could be classified as cohort event, whereas study sample could be divided into more levels of BP to assess the dose-response effect or just the baseline systolic and diastolic BP could be entered as predictive factor for developing hypertension. Please update your analyses with methods suitable for cohort with long term of follow-up! | Our study is a cohort study following up individuals free of hypertension at baseline with the outcome of incident hypertension. As the follow-up time of every study participant was identical, Cox regression analysis (which takes into account variable follow-up times) will not be appropriate method.. All study participants were re-examined after 5- and 11 –years at the same time point. Thus, if they had hypertension at the follow-up examination, we would not know at which time point they developed hypertension. Using the same follow-up time (5 yrs, respectively 11 yrs) for Kaplan-Meier survival time would give inaccurate data on survival times for people who developed hypertension. Furthermore, follow-up times will be the same for each individual and lead to similar estimated as odds ratios. Therefore, we decided to use logistics regression models, a perfectly valid statistical method to calculate associations if survival times are not available.
We have addressed this in the limitations of the study section adding the following text:
“Moreover, we were not able to analyze the data using survival analysis as information on the individual times-to-event was not available for the study participants who developed the outcome event.”
We did not find any evidence of bias or interaction by participants’ sex. Therefore, we present data for sexes combined to increase the power of the study. We have added the following sentence to the statistical methods section:
“As no evidence of bias or interaction was found by participants’ sex, the data were analyzed combining men and women.”
We agree that another research question may be to assess the dose-response effect of both systolic and diastolic blood pressure on future risk of hypertension. As for clinical decision making in primary health-care the use of hypertension categories are more common and recommended by international guidelines, we prefer to analyze the associations in this manuscript according to BP categories. However, we will address the issue of dose-response in a future work of these data. Also, it is not possible to extend the length of this manuscript and to address this important topic in depth which would require several tables/figures and much more text for Introduction, methods, results and discussion sections. |
Page 12, line 298
Page 5, line 171
|

Reviewer 2 Report
1. It is based on an interesting issue.
2. The rationale for the introduction is good.
3. The methodology is appropriate.
4. The results are accurate and sound.
5. The discussion is discussed in detail.
6. The supporting information in the manuscript is reliable and validated.
7. The manuscript contains innovations.
Author Response
Please note that there were no queries for Reviewer 2
Reviewer 3 Report
This is an nteresting study describing the evolution f BP values in a general population sample of Mauritian Island.
It is not clear whether some individuals in the prehypertensive Group were treated or not , please clarify
Is the presence of diabetes a predictor of hypertension in prehypertensive individuals ? or after adjustment the statistical significance of increased risk is lost ?
Author Response
Response to Reviewer’s comments
Manuscript ID: ijerph-304588
Editor’s and Reviewer’s Comments
| Our Response
| Location of edits |
Reviewer #3 | ||
It is not clear whether some individuals in the pre-hypertensive group were treated or not , please clarify | Participants with prehypertension at baseline did not receive antihypertensive treatment. Any person treated with antihypertensive drugs at baseline were considered as hypertensive, and excluded from the cohort as indicated in the methods section.
We have added following sentences to the methodology section:
A systolic blood pressure of 120 to 139 mmHg or a diastolic blood pressure of between 80 and 89 mmHg and free-of anti-hypertensive medication was considered as prehypertension. | Page 3, line 104 |
Is the presence of diabetes a predictor of hypertension in pre-hypertensive individuals? or after adjustment the statistical significance of increased risk is lost ? | The association between diabetes mellitus and future hypertension was not statistically significant after adjustments for the covariates (Table 3). Therefore, it was eliminated from the model. Only covariates that were significantly associated with the outcome and improved the model remained in the adjusted logistic regression model. The significance of diabetes was lost in the multivariable model due to fact that some other factors were confounders between diabetes at baseline and hypertension incidence at follow-up. |

Round 2
Reviewer 1 Report
The authors improved the quality of the manuscript and explained the limitations and reasons, why survival analyses couldn’t be performed. This version of the manuscript can be accepted. My suggestion is to present these limitations at the end of the manuscript.